# The Effectiveness of Glutathione Redox Status as a Possible Tumor Marker in Colorectal Cancer

**DOI:** 10.3390/ijms22126183

**Published:** 2021-06-08

**Authors:** Delia Acevedo-León, Lidia Monzó-Beltrán, Segundo Ángel Gómez-Abril, Nuria Estañ-Capell, Natalia Camarasa-Lillo, Marisa Luisa Pérez-Ebri, Jorge Escandón-Álvarez, Eulalia Alonso-Iglesias, Marisa Luisa Santaolaria-Ayora, Araceli Carbonell-Moncho, Josep Ventura-Gayete, Luis Pla, Maria Carmen Martínez-Bisbal, Ramón Martínez-Máñez, Leticia Bagán-Debón, Aurora Viña-Almunia, M. Amparo Martínez-Santamaría, María Ruiz-Luque, Jorge Alonso-Fernández, Celia Bañuls, Guillermo Sáez

**Affiliations:** 1Servicio de Análisis Clínicos, Hospital Universitario Dr. Peset-FISABIO, 46017 Valencia, Spain; acevedo.delia@gmail.com (D.A.-L.); estany_nur@gva.es (N.E.-C.); marisasantaolaria@gmail.com (M.L.S.-A.); aracarbonell@gmail.com (A.C.-M.); josep.ventura@gmail.com (J.V.-G.); mariamparo.21@gmail.com (M.A.M.-S.); mariiia.94@hotmail.com (M.R.-L.); joorgeaf96@gmail.com (J.A.-F.); 2Departamento de Bioquímica y Biología Molecular, Facultad de Medicina y Odontotología-INCLIVA, Universidad de Valencia, 46010 Valencia, Spain; lidia.monzo@gmail.com (L.M.-B.); eulalia.alonso@uv.es (E.A.-I.); 3Servicio de Cirugía General y Aparato Digestivo, Hospital Universitario Dr. Peset-FISABIO, 46017 Valencia, Spain; sean99cartu@hotmail.com; 4Servicio de Anatomía Patológica, Hospital Universitario Dr. Peset-FISABIO, 46017 Valencia, Spain; ncamarasalillo@gmail.com (N.C.-L.); perez_marebr@gva.es (M.L.P.-E.); escandon_jor@gva.es (J.E.-Á.); 5Instituto Interuniversitario de Investigación de Reconocimiento Molecular y Desarrollo Tecnológico, Universitat Politècnica de València—Universitat de València, 46022 Valencia, Spain; plablascoluis@gmail.com (L.P.); carmen.martinez-bisbal@uv.es (M.C.M.-B.); rmaez@qim.upv.es (R.M.-M.); 6CIBER de Bioingeniería, Biomateriales y Nanomedicina (CIBER-BBN), 28029 Madrid, Spain; 7Unidad Mixta UPV-CIPF de Investigación en Mecanismos de Enfermedades y Nanomedicina, Universitat Politècnica de València, Centro de Investigación Príncipe Felipe, 46012 Valencia, Spain; 8Unidad Mixta de Investigación en Nanomedicina y Sensores, Universitat Politècnica de València—Instituto de Investigación Sanitaria La Fe, 46026 Valencia, Spain; 9Departamento de Química, Universitat Politècnica de València, 46022 Valencia, Spain; 10Departamento de Química Física, Universitat de València, Burjasot, 46100 Valencia, Spain; 11Departamento de Estomatología, Facultad de Medicina y Odontología-INCLIVA, 46010 Valencia, Spain; leticia.bagan@uv.es; 12Centro de Salud San Isidro, Consorcio Hospital General Universitario de Valencia, 46014 Valencia, Spain; auroraalmunia@gmail.com; 13Servicio de Endocrinología y Nutrición, Hospital Universitario Dr. Peset-FISABIO, 46017 Valencia, Spain

**Keywords:** colorectal cancer, oxidative stress, GSH, GSSG, GSSG/GSH redox state, tumor markers

## Abstract

The role of oxidative stress (OS) in cancer is a matter of great interest due to the implication of reactive oxygen species (ROS) and their oxidation products in the initiation of tumorigenesis, its progression, and metastatic dissemination. Great efforts have been made to identify the mechanisms of ROS-induced carcinogenesis; however, the validation of OS byproducts as potential tumor markers (TMs) remains to be established. This interventional study included a total of 80 colorectal cancer (CRC) patients and 60 controls. By measuring reduced glutathione (GSH), its oxidized form (GSSG), and the glutathione redox state in terms of the GSSG/GSH ratio in the serum of CRC patients, we identified significant changes as compared to healthy subjects. These findings are compatible with the effectiveness of glutathione as a TM. The thiol redox state showed a significant increase towards oxidation in the CRC group and correlated significantly with both the tumor state and the clinical evolution. The sensitivity and specificity of serum glutathione levels are far above those of the classical TMs CEA and CA19.9. We conclude that the GSSG/GSH ratio is a simple assay which could be validated as a novel clinical TM for the diagnosis and monitoring of CRC.

## 1. Introduction

The role of oxidative stress (OS) in cancer has been a favored topic for research in the recent years. OS occurs when redox homoeostasis within the cell is altered. This imbalance may be due to either an overproduction of spontaneous reactive oxygen species (ROS) and/or a deficiency in antioxidant systems [1,2,3]. The deficiency of an antioxidant system will induce an accumulation of ROS within the cell. The inhibition of key enzymes involved in the synthesis of glutathione or ROS-scavenging enzymes cause sustained OS [3].

The microenvironment of solid tumors is a complex framework which is mainly characterized by enhanced inflammation and hypoxic status, which both lead to the generation of different cytokines and large amounts of hydrogen peroxide that contribute to the malignancy and dissemination of transformed cells [4,5,6]. Different studies have reported that tumor cells present a common OS profile that can be summarized by a decrease in antioxidant enzymes such as superoxide dismutase (SOD) and catalase (CAT) and an increase in glutathione peroxidase (GPx), accompanied by high levels of lipid peroxidation products and the oxidative DNA damage products such as 8-oxo-7,8-2′-dihydro-deoxiguanosine (8-oxo-dG) [7,8].

Globally, colorectal cancer (CRC) is the third most commonly diagnosed cancer in males and the second in females, according to the World Health Organization (WHO) Global Cancer Observatory (GLOBOCAN) database [9,10]. The incidence and mortality rates vary markedly around the world. Although survival in CRC patients has doubled over the past 20 years, partially due to the generalized administration of chemotherapy and a decrease in postsurgical mortality, about 1.93 million new cases were diagnosed and almost one million cancer deaths occurred in 2020 [11,12].

For the diagnosis and clinical monitoring of CRC, serum tumor markers (TMs) such as carcinoembryonic antigen (CEA) and the carbohydrate antigen recognized by the monoclonal antibody NS19.9 (CA 19.9) are currently used. However, their clinical usefulness remains controversial from diagnostic, prognostic, and surveillance points of view. In fact, data are insufficient to recommend the routine use of these markers (including serum CA19.9) in the management of patients with CRC [13]. Over the last two decades, the association between OS and CRC has been established, and a clear relationship has been found, demonstrating its profound influence on the progression of the disease [7,14,15].

The tripeptide L-γ-glutamyl-cysteinyl-glycine, or glutathione, is present in all mammalian tissues at 1–10 mM concentrations and exists in the thiol-reduced (GSH) and disulfide-oxidized (GSSG) forms [3]. GSH is the predominant form and accounts for >98% of total glutathione [16]. In its reduced state, GSH is the most abundant nonprotein sulfhydryl in the cell (ranging from 0.1 to 10 mM) and plays a very important role in maintaining cellular homeostasis and redox balance, representing an essential element of the intracellular defense against ROS. The depletion of GSH below a critical threshold is considered to be a marker of OS, which underlies the pathophysiology of a variety of different age-associated degenerative diseases including inflammatory and tumor processes [7,16,17,18].

Extensive research has focused on the role of the glutathione metabolic system in the development, diagnosis, and treatment of cancer [18,19]. A large number of studies have established an association between cancer incidence and various disorders of GSH-related enzyme functions. Alterations in glutathione S-transferases (GSTs) are some of the most frequently reported disorders [20]. However, although numerous studies have demonstrated a dysregulation of GSH in tumor tissues including CRC [7,8], less attention has been focused on blood GSH and GSSG/GSH levels. In fact, there is no compilation of clinical studies evaluating the role of glutathione itself and, at the present time, the information regarding the changes in the tripeptide at the systemic level and its use as a clinical biomarker is poor and inconclusive. With this work, we gain further insight into the effectiveness of glutathione in terms of its clinical validation and present experimental evidence which allow for the proposal of GSH and the percentage ratio GSSG/GSH as an emergent TM for the diagnosis and monitoring of CRC.

## 2. Results

Demographic, anthropometric, biochemical, and hematological parameters of controls and patients are listed in Table 1. The median and age ranges were 64.0 years (33–82) for the controls and 67.5 years (37–89) for the CRC patients.

No significant differences were observed between the two groups in terms of age and sex, but there were in terms of weight and height and consequently in body mass index (BMI). Therefore, and given that age also showed a trend, a univariate variance analysis was performed to compare all the studied markers, introducing BMI and age as covariates, in order to eliminate their possible confounding effect.

CRC patients showed significantly higher levels of glucose, albumin, transferrin, and IL-6, and lower total cholesterol, HDL-cholesterol, LDL-cholesterol, ferritin, iron, and transferrin saturation index than the control group. However, there were no differences regarding creatinine, urea, glomerular filtration, triglycerides, proteins, and C-reactive protein (CRP).

### 2.1. Histological Types of Diagnosed Tumors

The most frequent histological cancer types were adenocarcinomas (83.5%) followed by adenoma (10.1%) and to a lesser extent mutinous carcinoma (2.5%), mixed adenocarcinoma/mucinous carcinoma (1.3%), mixed mucinous carcinoma/ring cell carcinoma (1.3%) and gastrointestinal stromal tumors or GISTs (1.3%). Of the 80 CRC patients studied, 52 had colon (65%) and 28 rectal (35%) tumors.

The tumors were classified by the TNM system (applicable only to carcinomas) so the GIST tumor could not be classified. We regrouped these stages for the correlation study as indicated [21] into qualitative variables as follows—localized disease (stage 0): 44 patients (55.7%); progressive disease (stage 1): 26 patients (32.9%); and invasive disease (stage 2): 9 patients (11.4%).

### 2.2. Tumor Markers

As shown in Figure 1, significant differences for both CEA and CA 19.9 TM were observed between the control and the CRC groups. Twenty-four patients (30%) presented elevations of CEA and 14 CRC patients (17.5%) presented elevated levels of CA 19.9.

### 2.3. Serum Glutathione Levels

Serum glutathione levels were greatly affected in CRC patients as compared to control group. The concentration of GSH was reduced by more than 50%, while its oxidized form GSSG increased by more than 140%, which resulted in and was reflected by a very significant increase in the percentage ratio GSSG/GSH, indicating an important redox shift towards an oxidation state in the CRC patients (Figure 2).

### 2.4. Correlation between GSH, GSSG, and GSSG/GSH%, Tumor Markers, and Biochemical Parameters

The correlation of GSH, GSSG, and GSSG/GSH with CEA and CA 19.9 and with inflammatory parameters was studied in control and CRC patients (Table 2). The analysis showed that both CEA and CA 19.9 correlated positively with GSSG (r = 0.292 and r = 0.345, *p* < 0.001), and the GSSG/GSH ratio (r = 0.276 and 0.322, *p* < 0.001), and negatively with GSH (r = −0.270 and r = −0.292, *p* < 0.001), respectively. The analysis of inflammatory markers showed that both GSSG and the GSSG/GSH ratio correlated positively with IL-6, leukocytes, neutrophils, lymphocytes, neutrophil/lymphocyte index (N/L), platelets, and fibrinogen, and GSH correlated negatively with all of them. However, the correlation was moderate–low, with correlation coefficients lower than 0.500 in all cases but significant in all cases.

### 2.5. Glutathione Levels and Tumor Stages

The stages were categorized into three levels, as previously mentioned (localized disease = 0; progressive disease = 1; invasive disease = 2), with analysis using one-way ANOVA followed by the Student–Newman-Keuls (SNK) post hoc rank tests. Significant differences were found by stage in all the glutathione parameters (Figure 3). The differences for GSSG and GSSG/GSH% were significant at Stage 2.

Serum GSH levels were significantly reduced, while GSSG and the GSSG/GSH% ratio were significantly increased when comparing adenomas vs. carcinoma patients (data not shown).

### 2.6. Changes in Glutathione Levels after CRC Treatment

GSG, GSSG, and the GSSG/GSH% ratio tended to progressively recover during the first 12 months after treatment, with values close to those of the control group (Figure 4).

### 2.7. Evaluation of the Glutathione Levels: Contingency Tables and ROC Curves

The diagnostic tests (sensitivity, specificity, PPV, NPV, and accuracy) of the glutathione levels and TM were studied, and their respective receiver operating characteristic (ROC) curves were obtained. These cutoff points, with the results of the diagnostic tests, are shown in Table 3 and Figure 5.

The TM had maximum specificity and PPV, but the sensitivity was very low. The NPV and accuracy of CEA and CA 19.9 were below 60%. GSH presented maximum specificity and PPV, with a sensitivity of 78.8%. The GSSG/GSH% index presented values above 90% for all the diagnostic tests. The accuracy of the three markers ranged from 85.0% to 98.6%.

## 3. Discussion

The identification of easily determined biochemical molecules for their use as clinical markers of diseases continues to be a topic of great interest in translational research, especially when cancer is considered. This fact is especially important in the case of gastrointestinal tumors where there is a need to have sufficiently reproducible, sensitive, and specific markers that meet clinical expectations. The World Health Organization (WHO), in coordination with the United Nations and the International Labor Organization, defined a biomarker as “any substance, structure, or process that can be measured in the body or its products and influence or predict the incidence of outcome or disease” [22]. At the same time, the NIH defined this term as a “characteristic that is objectively measured and evaluated as an indicator of normal biological processes, pathogenic processes, or pharmacologic responses to a therapeutic intervention” [23]. To be a predictor of disease, a biomarker must be validated. Validation criteria include intrinsic qualities such as specificity, sensitivity, and knowledge of the confounding and modifying factors. In addition, the characteristics of the sampling and analytical procedures are of relevance when considering the constraints and non-invasiveness of sampling, stability of potential biomarkers, and the simplicity and speed of the analytical method.

The role of TMs includes their use as diagnostic markers, predictors of disease prognosis, and surveillance markers. The American Society for Clinical Oncology recommends for CRC that serum CEA testing be ordered preoperatively if it will assist in staging and surgical planning. Postoperative CEA levels should also be assessed every 3 months for stage II and III disease for at least 3 years if the patient is a potential candidate for surgery or chemotherapy for metastatic disease [13]. However, their clinical utility remains controversial from a diagnostic point of view, and the available data are still insufficient to recommend the routine use of other markers including serum CA 19.9 in the treatment of patients with CRC [13].

It is generally known that CRC is associated with OS through the imbalance in the oxidative/antioxidative state and DNA damage, which has been shown to underlie the pathogenic progression towards more advanced stages [7]. However, despite the pathogenic implications of OS [1,2,7,24,25,26,27,28,29], few OS markers are used routinely in the clinical setting, which may be for several reasons [16,29,30].

The association between oxidative/nitrosative stress and pathology is not always as clear as would be desirable due to certain recognized shortcomings of the methods used to measure OS/reactive nitrogen species (RNS) status in humans in vivo [2,16,29,31]. These shortcomings affecting the assays may be related to the limited specificity of the assay itself, the fact that the analyte being measured is not a specific ROS/RNS product, the lack of sufficient sensitivity to detect concentrations of the product in healthy individuals (thus not allowing the definition of a reference interval), the influence of external factors such as specific diet components, or the assay being too invasive for in vivo investigations in humans [16]. One OS byproduct which seems to overcome the above shortcoming is serum glutathione. However, the search with regard to the possible role of the tripeptide as a clinical marker and more specifically in cancer disease has not yet been carried out. GSH and its oxidation product GSSG together with its percentage redox ratio, GSSG/GSH%, is one of the most representative indicators of OS, and can be quantified in non-invasive biological samples with sufficient sensitivity using different analytical methods to detect its concentrations in both healthy individuals and pathological states [3,17,18,31,32,33,34,35,36].

In general, any condition associated with excessive ROS will decrease GSH levels or decrease the GSH/GSSG% ratio. Because it can be readily sampled, the oxidation/reduction status of blood glutathione is commonly used in investigations involving OS and free radical pathologies that occur in other tissues. Low GSH, high GSSG, and a lower GSH/GSSG% ratio have been found in blood from patients with various pathologies [37,38]. Measuring the GSH/GSSG ratio in pathological tissues and related experimental models is an excellent way to assess the potential efficacy of therapeutic strategies in maintaining cellular redox equilibrium [39].

In clinical studies, GSH and GSSG are most often measured in blood or in isolated red blood cells based on the assumption that, although indirect, this minimally invasive type of analysis provides valuable information on the redox balance of less accessible tissue and organs as well as the whole organism [40]. However, there is a high variability in the reported data for GSH and GSSG at the systemic level, which might reflect problems with the assays that are used. Consequently, measured concentrations of GSH and GSSG vary considerably between laboratories. Indeed, the results of some reviewed studies suggest that because large differences in GSH and particularly in GSSG exist even for control (healthy) individuals, an accurate revision of the methods used is required. However, a comparison of GSH levels analyzed using different assay methodologies and different biological samples is difficult to perform. Several methods have been used to determine GSH levels in biological samples [36]. At present, high-performance liquid chromatography (HPLC) and capillary electrophoresis are the most commonly used separation techniques to determine GSH and GSSG. However, these suffer from a lack of total automation and the high cost of the equipment [37].

GSH levels have been assessed in different gastrointestinal tumors such as pancreatic cancer [41,42,43] and gastric tumors [44]. Jagust et al. recently showed that glutathione metabolism plays an essential role in pancreatic cancer aggressiveness, supporting cancer stem cell survival, self-renewal, and chemoresistance [41]. Thus, cancer cells have increased glutathione levels to alleviate the effects of OS [42]. In this sense, other authors argue that excess GSH promotes tumor progression, where elevated levels correlate with increased metastasis [45]. Regarding CRC, there are various studies that associate OS with colorectal carcinogenesis [7,46,47]. In contrast to tissues, decreased serum levels of GSH have been observed in CRC patients as compared to control subjects [48,49,50,51] due to an increase after tumor treatment [52].

In line with these results, our study showed significant differences in the levels of both GSH and GSSG in the CRC group as compared to the healthy control group. GSH levels fell by more than 50%, while GSSG levels increased by more than 140%, leading to a very significant increase in the GSSG/GSH% ratio in the serum of CRC patients, indicating a clear change in the redox state of these CRC patients towards oxidation. Both the CEA and CA 19.9 TMs correlated positively with GSSG and GSSG/GSH% ratio and negatively with the serum concentration of GSH. GSH levels were significantly reduced, while GSSG and the glutathione redox state were significantly increased when comparing adenomas with carcinomas (data not shown). In accordance with other studies [53,54], a correlation analysis revealed a relationship between glutathione levels and inflammatory markers. The time course evolution of CRC patients showed that GSH, GSSG, and the GSSG/GSH% ratio tended to progressively recover during the first 12 months after treatment, with values close of those of the control group, although longer-term monitoring should be performed in order to learn whether the values completely normalize. Significant differences were detected even after the first month of treatment. Thus, the thiol redox state in CRC patients correlated significantly with both the tumor state and with the clinical evolution. However, our results indicate that GSSG/GSH levels increase with the tumor stage but are only significant in the most advanced stage (Stage 2) of the tumor. For usefulness as a reliable TM, significant changes should be detected in the initial stages of the disease. This could be related to the variance in the measurement of earlier tumor progression stages. It should be also emphasized that the implication of OS in cancer progression is not a unique mechanism in CRC, since other gastrointestinal diseases are also affected [41,42,43,44].

When we evaluated the diagnostic potential of the glutathione test based on its sensitivity and specificity, PPV, and NPV, together with the cutoff level and their comparison with CEA and CA 19.9, we found that the classic TMs had maximum specificity and PPV, but their sensitivity was very low. The NPV and the accuracy of both were below 60%. GSH presented maximum specificity and PPV with a sensitivity of 78.8%, while the GSSG/GSH% presented in all the diagnostic tests was above 90%. The accuracy of these three redox markers ranged from 87.1% to 97.1%. Statistical analysis showed a sensitivity and specificity of serum glutathione far above those of the classical TMs CEA and CA19.9, suggesting their usefulness for the diagnosis and monitoring of affected patients.

A strength of this study is that levels of GSH and GSSG as well as the GSSG/GSH% ratio was used not only to compare the baseline differences between control subjects versus CRC patients but also to monitor the disease based on tumor stage and disease evolution during a time course of one year after surgical treatment. In addition, the subjects were of a similar age, and both sexes were represented equally. On the other hand, we should point out some limitations. First of all, this was a pilot study with a modest number of subjects, and for this reason the statistical power was limited to compare different stages. Therefore, despite the close relationship between OS and carcinogenesis and the relevant role of the alterations of GSH levels in tumor diseases, its implication as a possible TM should be considered with caution, considering that the specificity of the relationship of GSSG/GSH ratio for CRC has not been evaluated in other gastrointestinal diseases and is efficiency as a clinical TM should be more widely studied. At present there are no studies with similar characteristics focused on investigating the clinical potential of glutathione based on its sensitivity, specificity, and accuracy as a TM.

In summary, the results presented suggest that serum levels of GSH and GSSG and the GSSG/GSH% ratio are candidates to be explored and may become effective TMs not only in CRC but possibly also for other types of tumors.

## 4. Materials and Methods

### 4.1. Study Design

This was a longitudinal and prospective, observational study in patients with a diagnosis of colorectal tumor who were candidates for tumor resection surgery and/or chemo-radiotherapy treatment and had been referred to the General Surgery and Digestive System Service of the University Hospital Dr. Peset. The follow-up period was of 1 year. A control group of age-matched healthy volunteers was included for a comparison with CRC patients at baseline.

This study was designed in accordance with the principles of ethics of the Declaration of Helsinki (Finland, 1964), and was evaluated and approved by the Clinical Research Ethics Committee of the University Hospital Dr. Peset. Informed consent was obtained from all subjects involved in the study.

### 4.2. Study Population

A total of 80 patients with a diagnosed colorectal tumor (both advanced adenomas and carcinomas) were incorporated into the study from March 2019 to January 2020. Advanced adenomatous neoplasia was considered when the polyps were 1 cm or more in diameter, with villous component or high-grade dysplasia [55].

The patients were selected based on the following criteria: patients of both genders without comorbidities who had not been treated with radio or chemotherapy before the surgery. The exclusion criteria in patients with CRC included any systemic or autoimmune diseases (diabetes, insulin resistance, hypertension, coronary heart disease, rheumatoid arthritis, and psoriasis), as well as lung, thyroid, liver, kidney, gastrointestinal and infectious diseases (chronic viral hepatitis and HIV infection). Additionally, smokers and patients who had taken drugs (antibiotics, non-steroidal anti-inflammatory drugs, glucocorticoids, vitamins and dietary supplements) within the preceding 3 months were excluded from the study.

Possible interferences at the time of the analysis of the selected metabolites due to nutritional factors were avoided through nutritional recommendations based on a Mediterranean diet and a dietary survey of all participants. A flow diagram of CRC patients throughout the study is shown in the Appendix A.

As a control population, 60 healthy subjects mostly aged around 50 years, of both genders, with a BMI <30 kg/m^2^, with clinical characteristics similar to those of the patients, and no clinical pathologies (especially dyslipidemia, diabetes mellitus, arterial hypertension, chronic renal failure, ischemic heart disease, or inflammatory bowel disease) were included in the present study.

In order to carry out the statistical studies of association of the different variables with the tumor staging, the AJCC TNM stages were transformed into the qualitative variables 0, 1, and 2 as follows: tumors located in the colon/rectum, including TNM stages 0, I, and II = Stage 0; regional tumors, affecting lymph nodes, TNM stage III = Stage 1; and advanced tumors with metastasis or invasion in sites distant from the peritoneum, TNM stage IV = Stage 2 [21].

### 4.3. Analytical Assays

#### 4.3.1. Biochemical and Hematological Studies in Serum Blood Samples

Serum samples were extracted after 12 h of fasting in dry 10 mL tubes with a silicone gel separator and coagulation accelerator. After clot retraction (about 30 min at room temperature) they were centrifuged at 3500× rpm for 5 min in a Rotina 380R Hettich centrifuge (Tuttlingen, Germany). Aliquots were separated from the serum for the determination of the GSH and GSSG, leaving a volume of 1 mL for the biochemical determinations, which were performed on the same day. The aliquots were frozen at −80 °C in the New Brunswick Scientific Premium U 410 freezer (Eppendorf, NJ, USA) until glutathione assay.

In each serum sample, a basic clinical analysis plus ferric metabolism, inflammation markers, and TMs was requested. Metabolites were analyzed in an automated chain of Architect C16000 equipment from Abbott (Chicago, Illinois, USA) and in the Cobas 600 from Roche Diagnostics (Manheim, Germany), following the spectrophotometric and immunochemical methodology of the manufacturer.

The analysis of biochemical parameters including triglycerides, total cholesterol and HDL cholesterol, glucose and insulin, total proteins, albumin, urea, creatinine, and electrolytes was performed by standard procedures on automatic analyzers. The validation of appropriate controls was performed each working day.

The glomerular filtration rate was estimated with the Chronic Kidney Disease Epidemiology Collaboration (CKD-EPI), a mathematical formula that includes serum creatinine, age, sex, and race as variables.

For the hematimetric analysis, samples of whole blood were taken in EDTA-K3 tubes and a Beckman-Coulter LH 500 hematology analyzer (Brea, CA, USA) was used. For fibrinogen analysis, samples of whole blood were taken in sodium citrate tubes and a ACL-TOP of Instrumentation Laboratory Company (Bedford, Massachusetts, USA) was employed.

#### 4.3.2. GSH, GSSG, and GSSG/GSH% Ratio

These values were analyzed with a glutathione reagent (Sigma-Aldrich, Article number 703002) using the specifications of Cayman’s glutathione assay based on a redox reaction between GSH and GSSG in the presence of GR with previous deproteinization of the sample to avoid the interference with protein sulfhydryl groups and further derivatization of GSH to GSSG with 2-vinylpyridine (Sigma-Aldrich, Article number 13229-2) in ethanol. The results of both are expressed in µmol/mL. The GSSG/GSH index was calculated by dividing both and multiplying by 100.

### 4.4. Statistical Analysis

The statistical analysis was performed with the Statistical Package for the Social Sciences (SPSS) program in its version 17.0 for Windows (Chicago, IL, USA). The results of the continuous quantitative variables are expressed as means and SD in tables and means and standard error of means or box and whiskers in figures, and the qualitative data in absolute numbers and percentages. The Kolmogorov–Smirnov test was used to assess normality for continuous variables, and the data conformed to normal distribution patterns. The differences in means between the control group and the CRC patients were compared using a Student’s t test for parametric samples or a Mann–Whitney U test for nonparametric samples. One-way ANOVA followed by a Student–Newman–Keuls post hoc test was used to compare more than 2 groups. Spearman’s correlation coefficients were employed to measure the strength of the association between glutathione levels (GSH, GSSG, and GSSG/GSH), CEA, and CA 19.9. Contingency tables and ROC curves were used to study the diagnostic characteristics of OS markers and TMs. A p-value <0.05 was considered statistically significant.

For the estimation of the sample size, the TMs CEA and CA 19.9 were chosen as reference variables and the means and standard deviation (SD) of serum samples from 30 healthy subjects and 30 patients with CRC were calculated. An α-type error of 5% and a statistical power of 80% for unpaired samples were fixed and calculated with the t-test of the statistical package R, version 2.14.2 (Auckland, NI, New Zealand). In the case of the CEA, for a mean difference of 5 ng/mL and a SD of 9.3 ng/mL, a sample size of 55 was obtained; in the case of CA 19.9, for a mean difference of 19.1 IU/mL and a DS of 36.2 IU/mL, the program obtained a result of 57 samples. For this reason, initially n = 60 was considered, but given that losses of between 10% and 20%were expected in the follow-up, the sample size was increased to a total of 80 patients.

## Figures and Tables

**Figure 1 ijms-22-06183-f001:**
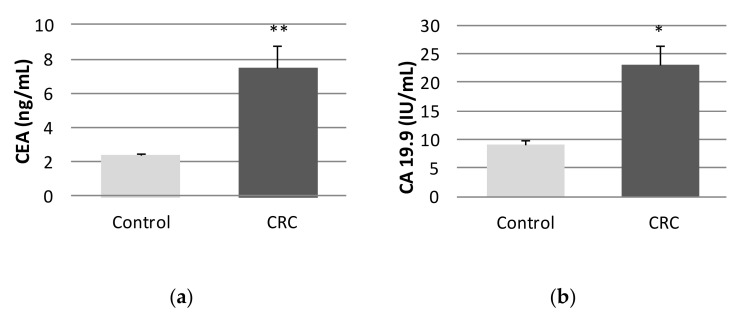
Tumor markers of controls and colorectal cancer (CRC) patients. (**a**) CEA and (**b**) CA 19.9. * *p*-value adjusted by age and body mass index (* *p* < 0.05; ** *p* < 0.01); Data are expressed as the mean ± standard error of the mean. CEA: carcinoembryonic antigen (normal value: <5 ng/mL); CA 19.9: carbohydrate antigen 19.9 (normal value: <37 U/mL).

**Figure 2 ijms-22-06183-f002:**
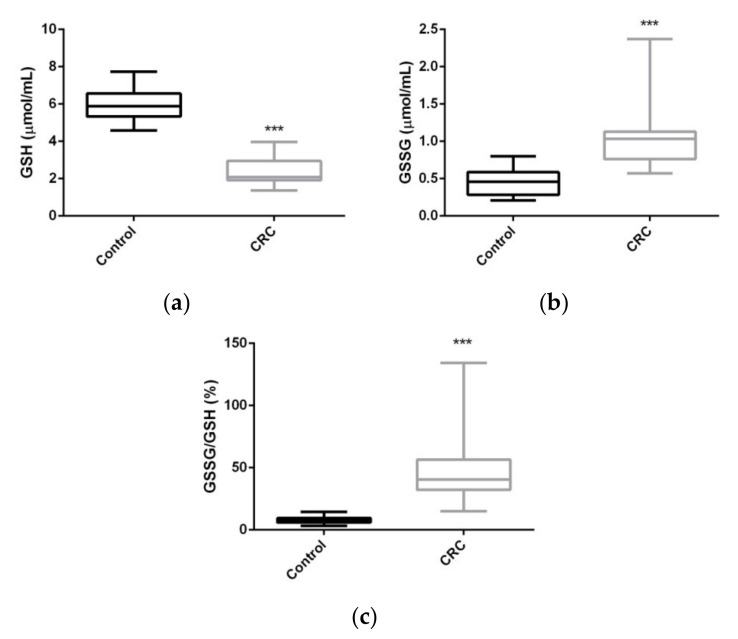
Levels of serum reduced glutathione (**a**), oxidized glutathione (**b**), and percentage ratio GSSG/GSH (**c**) in controls and CRC patients. *p*-value adjusted by age and body mass index (*** *p* < 0.001). GSH: reduced glutathione; GSSG: oxidized glutathione. Data are expressed as box and whiskers.

**Figure 3 ijms-22-06183-f003:**
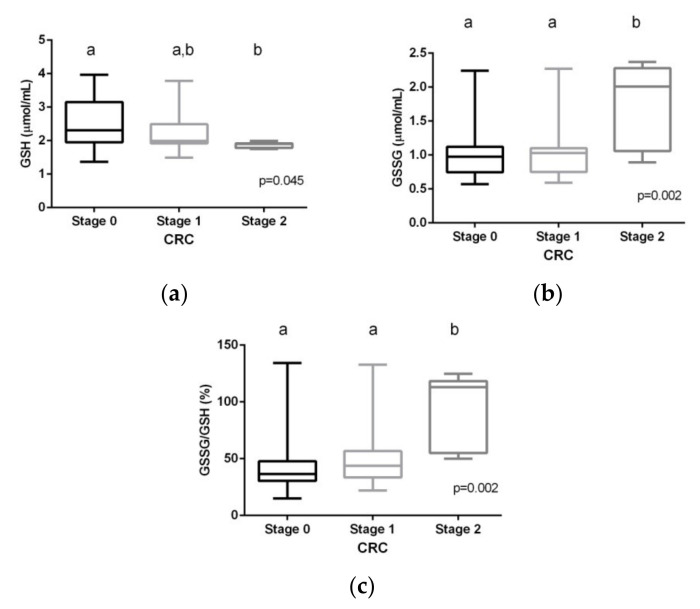
Reduced (**a**) and oxidized (**b**) glutathione levels and GSSG/GSH% ratio (**c**) grouped by tumor stages. Data are expressed as box and whiskers (stage 0, n = 44; stage 1, n = 26: stage 2, n = 9). Values with different superscript letters (a, b) were significantly different when the 3 groups were compared by one-way ANOVA followed by a Student–Newman-Keuls post hoc test. GSH: reduced glutathione; GSSG: oxidized glutathione.

**Figure 4 ijms-22-06183-f004:**
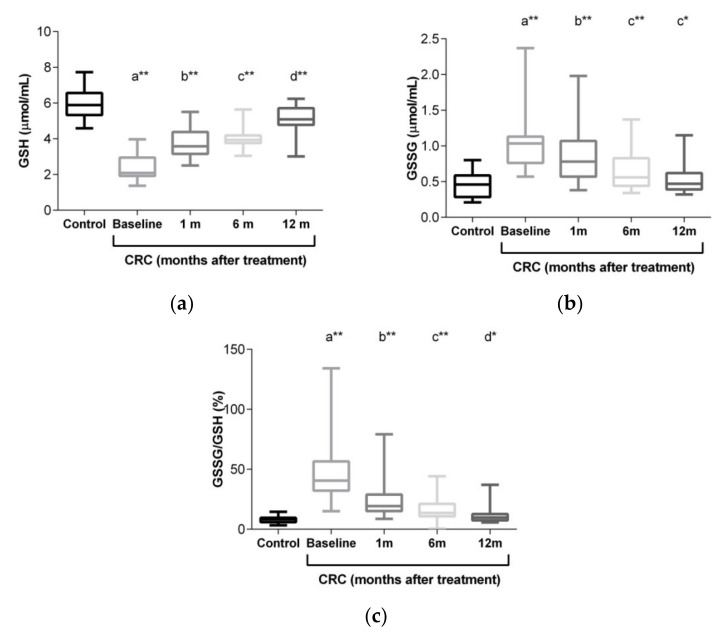
Time course evolution of serum glutathione levels after treatment of colorectal cancer (CRC) patients and controls. (**a**) Reduced glutathione (GSH); (**b**) oxidized glutathione (GSSG); (**c**) GSSG/GSH ratio. Data are expressed as box and whiskers. Values with different superscript letters (**a**–**d**) were significantly different when the evolutionary times of the CRC group were compared by repeated measures one-way ANOVA followed by a Student–Newman–Keuls post hoc test. * *p* < 0.05; ** *p* < 0.01 when control and CRC groups were compared with an unpaired Student’s *t* test.

**Figure 5 ijms-22-06183-f005:**
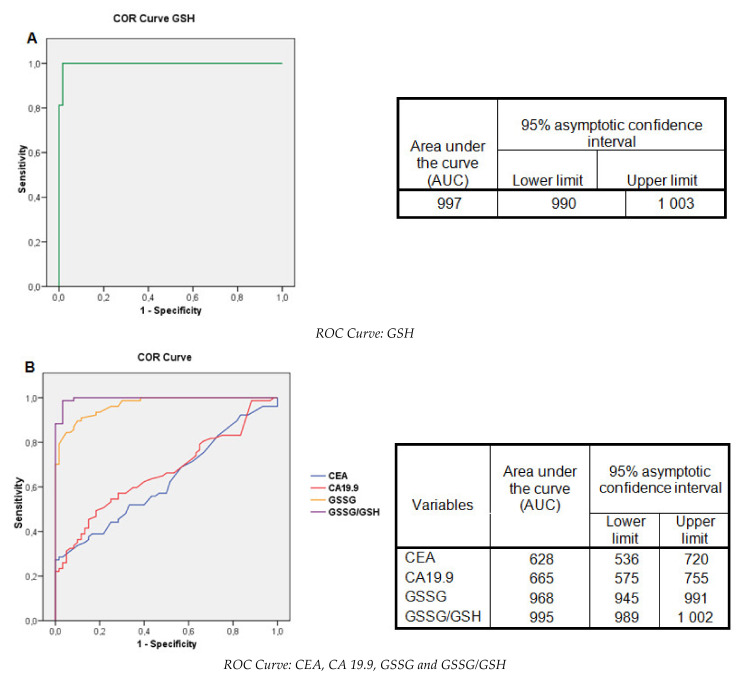
Receiver operating characteristic (ROC) curves for the analyzed markers. (**A**) Reduced glutathione (GSH); (**B**): CEA, CA 19.9, oxidized glutathione (GSSG), and the GSSG/GSH ratio.

**Table 1 ijms-22-06183-t001:** Demographic, anthropometric, biochemical, and hematological parameters of controls and colorectal cancer (CRC) patients.

Variable	Control (n = 60)	CRC (n = 80)	*p*-Value	* Adjusted *p*-Value
Age (years)	64.0 ± 9.0	67.5 ± 11.8	0.052	-
Male/Female (n; %)	36/24; 60/40	52/28; 65/35	0.548	-
Weight (kg)	74.4 ± 16.3	77.3 ± 15	<0.001	-
Height (cm)	168 ± 11	165.5 ± 9.8	<0.001	-
BMI (kg/m^2^)	26.1 ± 3.0	28.1 ± 3.9	0.001	-
Glucose (mg/dL)	96.2 ± 14.4	116.6 ± 52.3	<0.001	0.001
Creatinine (mg/dL)	0.9 ± 0.2	2.0 ± 8.6	0.319	0.269
Urea (mg/dL)	40.9 ± 7.2	38.8 ± 15.9	0.878	0.296
EGF (mL/min)	81.1 ± 8.7	78.9 ± 20.9	0.399	0.720
Total cholesterol (mg/dL)	195.7 ± 34.3	180.4 ± 39.1	0.018	0.026
HDL cholesterol (mg/dL)	50.7 ± 12.8	43.2 ± 10.8	<0.001	<0.001
LDL cholesterol (mg/dL)	144.9 ± 30.0	114.5 ± 34.7	<0.001	<0.001
Triglycerides (mg/dL)	112.0 (98;142.8)	108.5 (83.3;141)	0.954	0.777
Uric acid (mg/dL)	4.5 ± 1.6	5.3 ± 1.7	0.003	0.058
Albumin (g/dL)	3.9 ± 0.4	4.2 ± 0.5	0.001	<0.001
Total proteins (g/dL)	7.0 ± 0.5	6.9 ± 0.4	0.472	0.577
Ferritin (µg/L)	133.5 ± 75.4	67.5 ± 144.9	0.002	0.008
Iron (µg/dL)	79.7 ± 19.1	57.6 ± 41.0	<0.001	<0.001
Transferrin (mg/dL)	269.3 ± 46.5	291.1 ± 51.4	0.011	0.016
TSI (%)	30.5 ± 8.9	16.2 ± 11.1	<0.001	<0.001
CRP (mg/L)	6.2 ± 1.3	11.0 ± 23.6	0.072	0.076
IL-6 (pg/mL)	2.7 ± 1.4	19.7 ± 26.5	<0.001	<0.001
Leukocytes (x10^3^/mm^3^)	7.0 ± 1.7	7.7 ± 1.8	0.016	0.018
Neutrophils (%)	56.0 ± 6.0	62.9 ± 8.4	<0.001	<0.001
Lymphocytes (%)	29.8 ± 10.2	27.7 ± 12.0	0.221	0.531
N/L (-)	2.1 ± 1.4	2.7 ± 1.6	0.005	0.013
Platelets (×10^5^/mm^3^)	206.0 ± 60.0	253.5 ± 74.0	<0.001	<0.001
Fibrinogen (mg/dL)	352.3 ± 70.8	483.8 ± 101.7	<0.001	<0.001

* *p*-value adjusted for age and body mass index; n: number of cases; BMI: body mass index; EGF: estimated glomerular filtration; TSI: transferrin saturation index; CRP: C-reactive protein; IL-6: interleukin 6; N/L: neutrophil/lymphocyte index. The data are expressed as mean ± standard deviation. For the values that do not follow a normal distribution (triglycerides) the median (quartile 25/75) is used.

**Table 2 ijms-22-06183-t002:** Correlation between serum glutathione levels and GSSG/GSH (%), tumor markers, and biochemical parameters in controls and CRC patients.

	GSH (μmol/mL)	GSSG (μmol/mL)	GSSG/GSH (%)
Tumor markers			
CEA (ng/mL)	−0.270 **	0.292 **	0.276 **
CA 19.9 (IU/mL)	−0.292 **	0.345 ***	0.322 ***
Inflammatory markers			
CRP (mg/L)	n.s.	n.s.	n.s.
IL-6 (pg/mL)	−0.328 ***	0.419 ***	0.385 ***
Leukocytes (×10^3^/mm^3^)	n.s.	0.186 *	0.173 *
Neutrophils (%)	−0.318 ***	0.362 ***	0.330 ***
Lymphocytes (%)	0.226 **	−0.280 **	−0.240 **
N/L (-)	−0.175 *	0.232 **	0.181 *
Platelets (×10^5^/mm^3^)	−0.300 ***	0.240 **	0.239 **
Fibrinogen (mg/dL)	−0.462 ***	0.521 ***	0.552 ***

Data are expressed as Spearman’s correlation coefficient (r) with statistical significance (* *p* < 0.05; ** *p* < 0.01; *** *p* < 0.001) for each pair of variables. When the correlation is not significant, it is represented as n.s. GSH: reduced glutathione; GSSG: oxidized glutathione; CRP: C-reactive protein; IL-6: interleukin 6; N/L: neutrophil/lymphocyte index.

**Table 3 ijms-22-06183-t003:** Diagnostic tests of serum tumor markers (CEA and CA 19.9) and glutathione levels.

Marker	Cut off	S (%)	SP (%)	PPV ‡ (%)	NPV ‡ (%)	Accuracy (%)
CEA (ng/mL)	4.95	26.3	100	100	50.4	57.8
CA 19.9 (IU/mL)	40.0	17.5	100	100	47.6	52.9
GSH (μmol/mL)	3.17 *	78.8	100	100	77.9	87.9
GSSG (μmol/mL)	0.73	75.0	98.3	98.4	74.7	85.0
GSSG/GSH (%)	14.3	98.8	98.3	92.8	98.3	98.6

‡ Values calculated for a prevalence of the disease in the studied population of 0.57. * The result below this cutoff point is considered a positive test; the remaining markers are considered positive above their cutoff point. S: sensitivity; SP: specificity; PPV: positive predictive value; NPV: negative predictive value; GSH: reduced glutathione; GSSG: oxidized glutathione.

## Data Availability

The data presented in this study are available on request from the corresponding author.

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
