# Peer review of "The Effectiveness of Glutathione Redox Status as a Possible Tumor Marker in Colorectal Cancer"

_ijms, 2021, doi:10.3390/ijms22126183_

Round 1
Reviewer 1 Report
The authors present evidences of increased proportion of the oxidized glutathione/reduced glutathione percentage in colorectal cancer (CRC). The sample population analysed seems to be big enough, and ROC curve results show promising diagnosis values of GSSG/GSH ratio for CRC. In my opinion, the research is well conducted but I have missed some analysis that could be performed with all the data the authors already have. Also, the conclusions are very limited since the unbalanced levels of glutathione have not been analysed in other gastrointestinal tumours and the differences are significant in advanced CRC.
Introduction
The information provided and references used in the introduction are, in general, correct. Nevertheless, the references related to OS in cancer, and the role of glutathione in tumour tissues, are not very recent. Thus, I would encourage the author to offer a more up-to-date view of the interest of analysing the glutathione levels in CRC.
Results
As a general consideration, the experimental design is well planed but some analyses are missed. In order to consider this research as a pilot study to describe a new TM, the authors should certify that the sample size is big enough to drive to conclusions (authors can refer to Thabane et al. BMC Medical Research Methodology 2010, 10:1 http://www.biomedcentral.com/1471-2288/10/1).
Also, since some biochemical parameters are different between Control and CRC, it should be analysed whether there is also a correlation between glutathione levels and this parameters that could explain the differences.
Considering that GSSG/GSH ratio is significantly different at the stage 2 of the disease, it would be interesting to assess the ratio sensitivity, specificity and accuracy at different stages, in order to assess this new TM utility in early stages of the disease.
As a minor comments:
- Blox plots are a better way to present the data.
- Figure 1 should have proper Y-axis labels.
- Figure 2a also lacks of a proper Y-axis label.
- In figure 4, control group do not present any letter indicating signification. I understand it was included in the statistical analysis since the authors conclude that the glutathione levels go back to the control ones after the treatment.
Discussion and conclusions
The discussion section broadly addresses the relationship of the OS and glutathione levels with the physiological state of the patient, as well as the problems for an automated quantification of GSH levels. Instead, the authors limit themselves to summarizing the results that lead to the main conclusion (and which is presented as the title of the article) without going into detail about the limitations of their own study.
As indicated by the authors in the introduction, OS is a phenomenon occurring in cancer progression in general, not only in CRC but also in other gastrointestinal diseases (10.4252/wjsc.v12.i11.1410, 10.3390/ijms22041534, 10.3390/ijms21218113). Thus, any correlation between glutathione levels and CRC should be cautiously taken. The authors already indicate a correlation with various pathologies (ref. 64-66,69) and indicate that could also be useful as TM for other types of tumours. All these factors should be discussed deeper considering that the GSSG/GSH ratio specificity for CRC is not well demonstrated since no other diseases have been included in the study. Moreover, the differences are significant in advanced stages of the disease (stage 2); but a reliable TM would be more useful at initial stages. This should be also discussed properly.
Materials and methods
In general, the methods are well described, although authors should assess normality and equal variances before performing the ANOVA test.
References
More than half the references are from before 2010. The authors may want to update the information about the state of the art.
Author Response
Please, find attached the author's note to reviewer

Reviewer 2 Report
I consider subject manuscript in a form well prepared for publication. Authors presented the purpose of the work clear to all readers and the state of the research field was reviewed with publications published twenty or thirty years ago, but also by more recent ones. Although numerous challenges exist in translating biomarker research into the clinical space, "the measurement of GSH, GSSG, and GSH/GSSG% ratio in blood can be considered an index of the redox status in the whole-organism and a useful marker of diseases in humans". Authors mentioned limitations of their study in a sense that there is a high variability in the reported data for GSH and GSSG at systemic level. That is why every study in this area is a contribution to adaptation of standard protocols for cancer screening/detection/diagnosis.
Author Response
Suggestions and Comments for Authors
I consider subject manuscript in a form well prepared for publication. Authors presented the purpose of the work clear to all readers and the state of the research field was reviewed with publications published twenty or thirty years ago, but also by more recent ones. Although numerous challenges exist in translating biomarker research into the clinical space, "the measurement of GSH, GSSG, and GSH/GSSG% ratio in blood can be considered an index of the redox status in the whole-organism and a useful marker of diseases in humans". Authors mentioned limitations of their study in a sense that there is a high variability in the reported data for GSH and GSSG at systemic level. That is why every study in this area is a contribution to adaptation of standard protocols for cancer screening/detection/diagnosis.
Response: We are grateful to the reviewer for his/her evaluation of the manuscript and thank him/her for considering that our study is well prepared and objectives have been clearly presented.
We have revised the manuscript and made the corresponding changes in accordance with the reviewer’s comments.
Round 2
Reviewer 1 Report
The authors present a well-revised version of the paper. My concerns have been addressed properly. As a minor concern, I suggest a more extensive English revision, though. Nevertheless, I appreciate the big efforts from the authors to greatly improve the final discussion.